# A Piezoelectric and Electromagnetic Hybrid Galloping Energy Harvester with the Magnet Embedded in the Bluff Body

**DOI:** 10.3390/mi12060626

**Published:** 2021-05-28

**Authors:** Xia Li, Cheng Bi, Zhiyuan Li, Benxue Liu, Tingting Wang, Sanchuan Zhang

**Affiliations:** School of Mechanical and Power Engineering, Zhengzhou University, Zhengzhou 450001, China; cheng2106366223@163.com (C.B.); ggyuan11@163.com (Z.L.); wangtt97@126.com (T.W.); sc.zhang@zzu.edu.cn (S.Z.)

**Keywords:** piezoelectric, electromagnetic, hybrid energy harvester, galloping, wind-induced vibration

## Abstract

To meet the needs of low-power microelectronic devices for on-site self-supply energy, a galloping piezoelectric–electromagnetic energy harvester (GPEEH) is proposed. It consists of a galloping piezoelectric energy harvester (GPEH) and an electromagnetic energy harvester (EEH), which is installed inside the bluff body of the GPEH. The vibration at the end of the GPEH cantilever drives the magnet to vibrate, so that electromagnetic energy can be captured by cutting off the induced magnetic field lines. The coupling structure is a two-degree-of-freedom motion, which improves the output power of the energy harvester. Based on Hamilton’s variational principle and quasi-static hypothesis, the piezoelectric–electromagnetic vibrated coupling equation is established, and the output characteristics of GPEEH are obtained by the method of numerical simulation. Using the method of numerical simulation, studies a series of parameters on the output performance. when the wind speed is 9 m/s, the effective output power of the GPEEH is compared with the classical galloping piezoelectric energy harvester (CGPEH) who is no magnet. It is found that the output power of GPEEH 121% higher than the output power of CGPEH. Finally, set up an experimental platform, and test and verify. The experimental analysis results show that the simulated output parameter curves are basically consistent with the experimental drawing curves. In addition, when the wind speed is 9 m/s, under the same parameters, the effective output power of the GPEEH is 112.5% higher than that of the CGPEH. The correctness of the model is verified.

## 1. Introduction

With the popularization of the application of microelectronic devices, the workload of battery replacement is increasing [1]. At the same time, waste batteries which were replaced will cause environmental pollution [2]. Many scholars began to study environmental energy harvesting devices. According to different working principles, environmental energy harvesting devices can be divided into piezoelectric [3], electromagnetic [4], electrostatic [5], thermoelectric [6], and triboelectric [7]. Among them, piezoelectric energy harvester (PEH) has a wider operating frequency range [8] and a higher energy density [9]. Additionally, performances of PEH are not affected by the external magnetic field, temperature, and other factors. In the meantime, it is suitable to supply the power of low-power microelectronic devices [10,11,12,13]. The energy conversion mechanism of PEH is to convert the vibration energy of the environmental vibration energy into the strain energy of piezoelectric ceramic [14]. Based on the positive piezoelectric effect, positive and negative charges are generated on the upper and lower surfaces of the piezoelectric plate [15], and then the voltage is output. Wind, as a clean and common energy source [16], has been used by some scholars as an energy source, called wind-induced vibrating energy harvester. It is usually the form of the cantilever beam, which is simple, low in experimental cost, and can test comfortably. Additionally, the piezoelectric energy harvester with the style of cantilever beam can amplify the vibrating displacement of piezoelectric ceramics [17,18] that which is more conducive to our research on the performance of the energy capture. Currently, wind-induced vibrating cantilever beam energy harvester mainly includes four modes: gallop [19], flutter [20], vortex-induced [21], and wake [22].

Galloping generally happens on a complex, irregular, and non-streamlined cross-sectional structure with edges a galloping energy harvester has usually a single degree of freedom, as the spring-damping-mass model [23]. Compared with the other mechanisms of vibration, its aerodynamic model is easy, relatively. In addition, vibration has the characteristics of large displacement and large deformation, a wide operating frequency range, and a more obvious energy capture effect. Many scholars who study galloping energy harvesters mainly focus on (1) The influence of structural parameters of beams and cross-sections for output characteristics [24]; (2) Modeling of galloping force analysis [25]; (3) Introduce nonlinear factors [26]; (4) Hybrid galloping energy harvester [27]. Zhao et al. [28] compared the single-degree-of-freedom (SDOF) model, single-mode Euler–Bernoulli distributed parameter model and multi-mode Euler–Bernoulli distributed parameter model. The experimental results showed that the distributed parameter models can be more reasonable. The SDOF model can better predict the starting wind speed of the structure and electrical and aeroelastic elastic behaviors of galloping energy harvesters, and it is simple and easy to obtain the electromechanical coupling term of the model. Javed et al. [29] studied the influence of different aerodynamic loads on galloping vibrating mechanics. Various expressions of galloping force were proposed in the article. Simultaneously, output differences caused by expression changes were analyzed and compared with galloping force experimental results. By comparison, it is proved that the selection of polynomial approximation coefficients may lead to changes in the type of bifurcation, tip displacement, and acquisition power amplitude. Tang et al. [30] applied Synchronous Charge Extraction (SCE) technology to the output circuit of piezoelectric energy harvesters, and studied its applicability to piezoelectric energy harvesters with different degrees of electromechanical coupling and PEH excited at different frequencies. The results showed that under non-resonant frequency or weakly coupled vibration, SCE technology can significantly improve energy harvesting efficiency. Dai et al. [31] introduced delay feedback into the nonlinear cantilever beam energy harvester, and analyzed the influence of delay time τ and linear coefficient k3 on the cut-in wind speed. The results showed that by lead to nonlinearity, the time delay control can effectively control the speed of vibration. Most scholars have improved the structure of the bluff body. In order to improve the response characteristics of the harvester, Alhadidi et al. [32]. proposed a bluff body structure with a Y-shaped fin. In the same configuration, the time for the Y-shaped fines energy harvester to reach stable cyclic oscillation is reduced by 75% than that of the non-Y-shaped energy harvester. Wang et al. [33] established the electromechanical coupling model for piezoelectric energy harvesters with the different vertex of triangular sections, studied the influence of electromechanical coupling on the behavior of energy harvesters, and expanded the aerodynamic empirical coefficient from 10–90° to 10–170°. Abdelkefi et al. [34] proposed a nonlinear parameterized model, to study the effect of the cross-sectional shapes of the bluff bodies for the output efficiency of the energy harvester. It is found that the output power of the energy harvester with the triangular section of the bluff body is the largest in low wind speed; when the wind speed is high, the power output power of the energy harvester with the D- section bluff body is the best. In addition, some scholars have studied the nonlinearity structure of beam, Binyet, Emmanuel Mbondo et al. [35] proposes a low-frequency undulating flexible plate placed in the wake of a square cylinder. Using water tunnel experiments, particle image velocimetry and fluid–structure interaction modeling. This result finds that the longer the board, the higher the output power and the lower the swing frequency.

In order to suffice higher energy supply demand and wider working conditions of energy supply devices, many scholars have been carrying out related studies for the direction of hybrid energy harvester [36,37]. Li et al. [38] proposed a piezoelectric and electromagnetic hybrid energy harvester based on the truss mechanism. The movement of the magnet changes the magnetic flux to achieve capture of electromagnetic energy, and at the same time drives the truss mechanism to move and stretches the piezoelectric ceramics makes it generate electricity. The experiment showed that the maximum peak output power of the energy harvester exceeds 0.33 W when it is excited by 0.7 g. Toyabur et al. [39] proposed a multi-degree-of-freedom, multi-vibration mode hybrid piezoelectric–electromagnetic energy harvester, which can work in four resonance modes in the range of 12–22 Hz. Under the 0.4 g of acceleration and the optimal load, the maximum output power of the energy harvester in the third resonance mode is 250.23 µW. Xu et al. [40] proposed a piezoelectric–electromagnetic nonlinear energy harvester with a spring-connected magnet at the end of the magnet, it has a higher performance of output characteristics than that of the ordinary energy harvesters. However, most of the studies only study a single degree of freedom, and there are few studies on galloping piezoelectric–electromagnetic coupling structures by using wind. Meanwhile, the coils of EEH are generally placed on the outside of the structure, which increases the space occupied by the energy harvester.

This paper presents a novel GPEEH whose coils are placed inside. It can make reasonable use of space. First, it is introduced for the working principle of GPEEH. Second, based on Hamilton’s variational principle and the Quasi-static hypothesis derives the coupled vibration equation of the structure. Third, the influence of relevant parameters on the output characteristics of the energy harvester is studied by numerical simulation. Finally, there are some experimental researches to verify the correctness of the model. 

## 2. Working Principle and Theoretical Modeling

### 2.1. Design and Working Principle

The GPEEH is made up of a galloping piezoelectric energy harvester (GPEH) and an electromagnetic energy harvester (EEH) which is shown in Figure 1. It mainly contains a copper cantilever beam, a PZT-5H, and a hollow bluff body. The piezoelectric sheet is bonded at the fixed end of the beam and the bluff body is installed on another free end of the beam. When the stable wind flows through the bluff body, the negative aerodynamic damping due to flow separation and vortex shedding will cause the wind-induced galloping instability of the cantilever beam, resulting in ultimate cycle oscillation.

The EEH is installed inside the bluff body and mainly includes a base, a spring, a guide column, a magnet, a coil, and a cylinder. When the bluff body is driven by aerodynamic force to generate excitation, EEH swings with it. 

In this structure, the displacement of the cantilever beam and displacement of the magnet are coupled with each other (two degrees of freedom). Due to the output power of GPEH and EEH are different, GPEEH can supply power to two external loads at the same time and can also be combined for power supply. The combination of two energy harvesting methods achieves the purpose of improving energy harvesting efficiency. Simultaneously, GPEEH can improve the space utilization rate of the energy harvester. 

Working principle of GPEEH: on the one hand, wind flows through the bluff body from the positive direction of the z-axis. When the wind speed reaches the start-up wind speed of this structure, the aerodynamic forces at the y-axis direction component generated to drive the bluff body to make a stable cyclic vibrated movement which causes PZT-5H to deform. Simultaneously, the positive piezoelectric effect causes a potential difference between the upper and lower surfaces of the piezoelectric sheet, supply power to load *R*_1_ through wires. On the other hand, the swing of the bluff body promotes the magnet to move in the direction of the guide column. The displacement is related to the vibrated behavior of the end of the cantilever beam, so the transverse displacement of the cantilever beam and the displacement of the magnet along the guide column direction are coupled with each other. The vibrating magnets hold the movement of cutting the magnetic induction line in the coil. According to Faraday’s law of electromagnetic induction, there will be currently generated in the coil, which can supply power to the load *R*_2_.

The cross-sectional view is shown in Figure 2. It is found that the cantilever beam is a stepped plate, which prevents PZT-5H from breaking. Explain the symbols in the picture: *h*_1_, *h, h*_p_ are the thickness of the front and hind step of the cantilever beam, and the thickness of the piezoelectric sheet, respectively. *L*_b_, *L*_s_, *L*_1_, *L*_2_ are the total length of the cantilever beam and the bluff body, the distance between the fixed end and the left, right faces of the piezoelectric sheet, and, respectively. *b*_s_ is the width of the bluff body, and the distance from the top surface of the bluff body to the bottom surface of the magnet is *L*_k_, which contains the free extension of spring *L*_k0_ and spring displacement change during the balance $\frac{M_{2}g}{K_{2}}$. *R*_o_, *R*_A_ is outside and inside the radius of the cylinder, respectively. 

### 2.2. Theoretical Modeling

In order to build the model of GPEEH in theory, the following assumptions need to be made:(a)Sticky pieces of the structure are fixed and bonded, ignoring the influence of the adhesive for the structure;(b)Piezoelectric cantilever beam agrees Euler–Bernoulli beam theory;(c)An average and stable wind blows along the positive direction of the z-axis;(d)Treat the magnet as an ideal magnetic dipole;(e)Assume absolute smoothness between the guide rod and the magnet bearing.(f)When the structure is vibrating, it agrees to the quasi-static assumption.

Under the above assumptions, the expression of the system vibration equation derived via the generalized Hamilton’s principle [41]:(1)$$\int_{t_{1}}^{t_{2}}\left[\delta \left(T_{p}-U_{p}\right)+\delta W_{p}\right]dt=0\;$$
where $T_{p}$, $U_{p}$, $W_{p}$ is the total kinetic energy, the total potential energy, and the virtual work of the system.

The deformation diagram of the cantilever beam is shown in Figure 3. The fixed end of the cantilever beam is set as the origin of coordinates, the direction along with the initial position and the direction perpendicular to the x-axis is set as the y-axis direction. The angle between the end of the cantilever beam and the horizontal plane is $\beta \approx w^{\prime }\left(L_{b},t\right)$, $w\left(L_{b},t\right)$ represents the displacement of the end of the cantilever beam along the y-axis, and *u*(t) represents the relative displacement of the spring relative to the free extension position during the vibration of the magnet.

### 2.3. Kinetic Energy, Total Potential Energy and Virtual Work

The kinetic energy of this model consists of *T*_A_ (the kinetic energy of the cantilever beam) and *T*_B_ (the sum of the kinetic energy of the bluff body and the kinetic energy of components of the EEH). They can be solved by Equations (2) and (3):(2)$$\begin{array}{l} T_{A}=\frac{1}{2}A_{1}\rho \int_{0}^{L_{2}}{\dot{w}}^{2}\left(x,t\right)dx+\frac{1}{2}A\rho \int_{L_{2}}^{L_{b}}{\dot{w}}^{2}\left(x,t\right)dx\\ +\frac{1}{2}A_{p}\rho_{p}\int_{L_{1}}^{L_{2}}{\dot{w}}^{2}\left(x,t\right)dx \end{array}$$
(3)$$\begin{array}{ll} T_{B}= & \frac{1}{2}M_{2}{\left[\dot{w}\left(L_{b^{,}}t\right)+\left(L_{k}+u\left(t\right)\right){\dot{w}}^{\prime }\left(L_{b},t\right)\right]}^{2}\\ & +\frac{1}{2}M_{2}{\dot{u}}^{2}\left(t\right)+\frac{1}{2}m_{d}\dot{w}{\left(L_{b},t\right)}^{2}\\ & {+\frac{1}{2}m_{g}\left[\dot{w}\left(L_{b},t\right)+L_{g}{\dot{w}}^{\prime }\left(L_{b},t\right)\right]}^{2}\\ & {+\frac{1}{2}m_{x}\left[\dot{w}\left(L_{b},t\right)+L_{x}{\dot{w}}^{\prime }\left(L_{b},t\right)\right]}^{2}\\ & +\frac{1}{2}I_{g}{{\dot{w}}^{\prime }}^{2}\left(L_{b},t\right)+\frac{1}{2}I_{x}{{\dot{w}}^{\prime }}^{2}\left(L_{b},t\right)\\ & {+\frac{1}{2}m_{t}\left[\dot{w}\left(L_{b},t\right)+L_{t}{\dot{w}}^{\prime }\left(L_{b},t\right)\right]}^{2}\\ & +\frac{1}{2}I_{\mathrm{tip}}{{\dot{w}}^{\prime }}^{2}\left(L_{b},t\right)+\frac{1}{2}I_{t}{{\dot{w}}^{\prime }}^{2}\left(L_{b},t\right)\\ & {+\frac{1}{2}m_{\mathrm{tip}}\left[\dot{w}\left(L_{b},t\right)+L_{\mathrm{tip}}{\dot{w}}^{\prime }\left(L_{b},t\right)\right]}^{2} \end{array}$$
where *A*, *A*_1_, *A*_p_ is the cross-sectional area of the front section and hind section of the stepped substrate, and the cross-sectional area of PZT-5H, respectively. Additionally, $\rho ,\rho_{p}$ is the density of copper and PZT-5H. $w\left(x,t\right)$ is the transverse displacement of the cantilever beam substrate, $u\left(t\right)$ represents the displacement of the magnet. $\dot{w}\left(x,t\right)$ and $w^{\prime }\left(x,t\right)$ are, respectively the linear velocity of the particle with the distance x from the fixed end of the cantilever beam, and the angle between the tangent direction of the cantilever beam passing through the particle and the X axis when the time is t. $\dot{w}\left(x,t\right)=\frac{\partial w\left(x,t\right)}{\partial t}$, $w^{\prime }\left(x,t\right)=\frac{\partial w\left(x,t\right)}{\partial x}.M_{2}$ is the magnet mass. $\dot{w}\left(L_{b},t\right)$ is the velocity at $x=L_{b}$ of the cantilever beam. ${\dot{w}}^{\prime }\left(L_{b},t\right)$ represents the angular velocity at the end of the cantilever beam. *m*_g_, *m*_x_, *m*_t_, *m*_tip_, *I*_g_, *I*_x_, *I*_t_, *I*_tip_ stand for the mass and moment of inertia of guide column, coil, cylinders, and bluff body. The total kinetic energy of the system is:(4)$$T_{p}=T_{A}+T_{B}$$

The total potential energy of the system includes the elastic potential energy of the cantilever beam$\;U_{c}$, the elastic potential energy of the spring$\;W_{k}$, the electric potential energy$\;W_{e}$, the electromagnetic energy$\;W_{m}$, and the gravitational potential energy$\;W_{g}\;$of the bluff body and its built-in components:(5)$$U_{c}=\frac{1}{2}\int_{V}\varepsilon^{2}EdV+\frac{1}{2}\int_{V_{p}}\varepsilon T_{s}dV_{p}$$
(6)$$W_{m}=\frac{1}{2}L_{c}I_{c}{\left(t\right)}^{2}$$
(7)$$W_{k}=\frac{1}{2}K_{2}u{\left(t\right)}^{2}$$
(8)$$\&\;W_{e}=\frac{1}{2}\int_{V_{p}}E_{3}DdV-\theta_{e}\dot{Q}\left(t\right)u\left(t\right)$$
(9)$$\begin{array}{l} W_{g}=m_{g}g\left[w\left(L_{b},t\right)+L_{g}w^{\prime }\left(L_{b},t\right)\right]\\ +m_{x}g\left[w\left(L_{b},t\right)+L_{x}w^{\prime }\left(L_{b},t\right)\right]\\ +m_{t}g\left[w\left(L_{b},t\right)+L_{t}w^{\prime }\left(L_{b},t\right)\right]\\ +m_{\mathrm{tip}}g\left[w\left(L_{b},t\right)+L_{\mathrm{tip}}w^{\prime }\left(L_{b},t\right)\right]\\ +M_{2}g\left[w\left(L_{b},t\right)+L_{k}w^{\prime }\left(L_{b},t\right)\right] \end{array}$$
where *V* is the volume of substrate, *V*_p_ is the volume of PZT-5H, *E* is Young’s modulus of copper, *L*_g_, *L*_x_, *L*_t_, *L*_tip_ are the height of the guide column, coil, cylinder, and bluff body, respectively.

In Equation (5), considering the characteristics of large deformation of galloping, ε of the strain can be expressed as [42]:(10)$$\varepsilon =-y\left(w^{\prime \prime }\left(x,t\right)+\frac{1}{2}w^{\prime \prime }\left(x,t\right)w\prime {\left(x,t\right)}^{2}\right)$$

This model satisfies mechanical clamping and electrical short circuit (the second type of boundary conditions of piezoelectricity), and the equation is as follows:(11)$$\;\left[\begin{array}{c} T_{s}\\ D \end{array}\right]=\left[\begin{array}{cc} c_{11}^{E} & -e_{31}\\ e_{31} & \varepsilon_{33}^{s} \end{array}\right]\left[\begin{array}{c} \varepsilon \\ E_{3} \end{array}\right]$$
where $T_{s}\;$is the stress of the cantilever beam, D is electrical displacement, $c_{11}^{E}$ is the elastic stiffness coefficient of PZT-5H,$\;e_{31}$ is the piezoelectric constant of the piezoelectric sheet,${\;\varepsilon }_{33}^{s}$ is the dielectric constant, $E_{3}=\frac{V\left(t\right)}{h_{p}}$ represents electric field intensity, and $V\left(t\right)$ is the voltage which generates by PZT-5H.

Put Equations (5)–(11) into Equation (12) to get the total potential energy $U_{p}$ of the system.
(12)$$\;U_{p}=U_{c}+W_{k}+W_{g}-W_{e}-W_{m}$$

The virtual work of the system Wp includes the$\;W_{R_{1}}$of virtual work which made by the load resistance *R*_1_,$\;W_{R_{2c}}$ of virtual work produced by the coil internal resistance *R*_c_ and load resistance *R*_2_, the $W_{\mathrm{cs}}$ of virtual work done working by the damping force of the piezoelectric cantilever. The $W_{\mathrm{kcs}}$ of virtual work is generated by spring damping force. Additionally, the virtual work is done by the $F_{y}$ of the aerodynamic force along the y-axis and $M_{y}$ of the moment of $F_{y}$.

The virtual work of the load resistance *R*_1_ can be expressed as:(13)$$\;\delta W_{R_{1}}=V\delta Q_{1}$$
(14)$$\;\delta W_{R_{2c}}=-\delta Q_{2}\left(R_{2}+R_{c}\right)I_{c}$$
(15)$$\;\delta W_{\mathrm{cs}}=\int_{0}^{L_{b}}C_{s}\dot{w}\left(x,t\right)\delta w\left(x,t\right)dx$$
(16)$$\;\delta W_{\mathrm{kcs}}=C_{s2}\dot{u}\left(t\right)\delta u\left(t\right)$$
where $V=R_{1}\frac{dQ_{1}}{dt}$, $Q_{1}$ is the charge generated by the piezoelectric sheet.$\;Q_{2}$ is the charge generated by the coil. *R*_c_ is coil internal resistance. $I_{c}\;$is the current generated by the coil. $C_{s2}$ is the mechanical damping of the piezoelectric cantilever beam and spring mechanical damping.

Then, the system virtual work is expressed as:(17)$$\;\delta W_{p}=-\delta W_{R_{1}}-\delta W_{R_{2c}}-\delta W_{\mathrm{cs}}-\delta W_{\mathrm{kcs}}+F_{y}\delta w\left(L,t\right)+M_{y}\delta w^{\prime }\left(L,t\right)$$

Putting Equations (13)–(16) into Equation (17) can get the system virtual work.

### 2.4. Aerodynamic Force and Aerodynamic Torque

The aerodynamic force generated by galloping can be broken down into three directions: x, y, and z. The driving force for the stable cyclic oscillation of the bluff body is mainly provided by the aerodynamic force $F_{y}$. We assume that the resultant force of the bluff body per unit length in the y-direction is $f_{y}$, then:(18)$$\;f_{y}=\frac{1}{2}\rho_{a}U^{2}b_{s}C_{y}$$

Aerodynamic coefficient *C*_y_ is related to
attack angle α of the bluff body and the experience coefficients$\;a_{1}\;$and$\;a_{3}$:(19)$$C_{y}=a_{1}tan\alpha +a_{3}{\left(tan\alpha \right)}^{3}$$

Because the bluff body is highly long, the transverse velocity along the length of the bluff body is quite different, and the aerodynamic attack angle α is considered as a function, which is expressed as the independent variable s of bluff body coordinates along the guide column, and the origin is at the center of the top surface, which is near the cantilever bluff body:(20)$$\;\alpha \left(s\right)=\tan^{-1}\left(\frac{\dot{w}\left(L,t\right)+s{\dot{w}}^{\prime }\left(L,t\right)}{U}\right)$$

The $F_{y}$ component of the aerodynamic force on the bluff body along the y-axis can be expressed as:(21)$$\;F_{y}=\int_{0}^{L_{\mathrm{tip}}}f_{y}\left(s\right)ds$$

Then, aerodynamic torque $M_{y}$ can be expressed as:(22)$$\;M_{y}=\int_{0}^{L_{\mathrm{tip}}}sf_{y}\left(s\right)ds$$

Putting Equations (18)–(20) into Equations (21) and (22) can get the aerodynamic force $F_{y}$ and its torque $M_{y}$.

### 2.5. Transverse Displacement Model Solution

Using the Galerkin procedure, the transverse displacement of the cantilever beam $w\left(x,t\right)$ can be expressed as [27]:(23)$$\;w\left(x,t\right)=\sum_{i=1}^{\infty }\varphi_{i}\left(x\right)q_{i}\left(t\right)$$
where $\varphi_{i}\left(x\right)$ is the modal shape of the cantilever beam, $q_{i}\left(t\right)$ is the modal coordinate, then and *i* is regarded as the modal order.

To determine which vibration mode dominates the cantilever beam vibration, this paper uses finite element software to perform modal analysis of GPEEH and the results are shown in Figure 4. It can be found that these four-order modes are bending vibration, torsional vibration, bending vibration where the cantilever beam has been deformed, and bending vibration along the width direction of the cantilever beam. As shown in Figure 4, the vibration frequencies of the first to fourth modes are 5.57873 Hz, 43.1367 Hz, 76.5378 Hz, and 233.78 Hz, respectively. The frequency of energy source captured by environmental energy is generally low, the first order vibration frequency of the energy harvesting structure is usually studied [43]. The first-order mode (bending vibration) is the dominant vibration mode of the primary energy trap.

Therefore, the transverse displacement can be simplified as:(24)$$w\left(x,t\right)=\varphi_{1}\left(x\right)q_{1}\left(t\right)$$

$\varphi \left(x\right)$ can be obtained by simulation [44] or boundary conditions combined with an undetermined coefficient method [45] to obtain more accurate results. This paper obtains the modal function by combining boundary conditions with the method of undetermined coefficients.

Since the piezoelectric sheet has little effect on the modal function, to simplify the fitting effect, the influence of the piezoelectric sheet on the model function is ignored, and the model shape can be regarded as:(25)$$\left\{\begin{array}{l} \varphi_{A}^{\prime }\left(x\right)=C_{a1}cos\left(\beta_{a}x\right)+C_{a2}sin\left(\beta_{a}x\right)+C_{a3}cosh\left(\beta_{a}x\right)+C_{a4}sinh\left(\beta_{a}x\right),x\in \left[L_{2},L_{b}\right]\\ \varphi_{B}^{\prime }\left(x\right)=C_{b1}cos\left(\beta_{b}x\right)+C_{b2}sin\left(\beta_{b}x\right)+C_{b3}cosh\left(\beta_{b}x\right)+C_{b4}sinh\left(\beta_{b}x\right),x\in \left[0,L_{2}\right] \end{array}\right.$$
$$\varphi_{B}\left(L_{2}\right)=\varphi_{A}\left(L_{2}\right),\varphi_{B}\prime \left(L_{2}\right)=\varphi_{A}\prime \left(L_{2}\right)\;D_{A}\varphi_{A}^{\prime \prime }\left(L_{2}\right)=D_{B}\varphi_{B}^{\prime \prime }\left(L_{2}\right),D_{A}\varphi_{A}\prime \prime \prime \left(L_{2}\right)=D_{B}\varphi_{B}\prime \prime \prime \left(L_{2}\right)\;EI_{3}\varphi_{A}^{\prime \prime }\left(L_{b}\right)-\omega_{1}{}^{2}I_{\mathrm{tip}}\varphi_{A}^{\prime }\left(L_{b}\right)-\omega_{1}{}^{2}M_{\mathrm{tip}}\frac{L_{\mathrm{tip}}}{2}\varphi_{A}\left(L_{b}\right)=0$$
(26)$$EI_{3}\varphi_{A}^{\prime \prime \prime }\left(L_{b}\right)+\omega_{1}{}^{2}M_{\mathrm{tip}}\varphi_{A}\left(L_{b}\right)+\omega_{1}{}^{2}M_{\mathrm{tip}}\frac{L_{\mathrm{tip}}}{2}\varphi_{A}^{\prime }\left(L_{b}\right)=0$$
where $\varphi_{A}^{\prime }\left(x\right)$ is the transverse displacement of the rear cantilever beam and transverse displacement of the front cantilever beam, respectively. $\omega_{1}$ is the 1st natural frequency of GPEEH.$\;m_{A},m_{B}$ are the mass per unit length of the first half and the rear half of the beam. $D_{A}=EI_{3}$, $D_{B}=EI_{1}$. $I_{1}$,$I_{3}$ is the moment of inertia of the front and rear of the step beam along the z-axis. $I_{1}=\frac{bh_{1}^{3}}{12}$, $I_{3}=\frac{bh^{3}}{12}$.$\;\beta_{a}=\sqrt{\frac{\omega_{1}}{D_{A}/m_{A}}}$, $\beta_{b}=\sqrt{\frac{\omega_{1}}{D_{B}/m_{b}}}$.

According to the corresponding displacement, rotation angle, layered position, bending moment, and continuous conditions of shear force, it can be defined: Equation (26) can be used to solve the coefficients $C_{aj}$,$C_{bj}$ in Equation (25). Normalize the known mode function to obtain the lateral displacement $w\left(x,t\right)$.

Arrange Equations (2)–(12) to get kinetic energy $T_{p}$ and potential energy $U_{p}$, then the Lagrange function $L=T_{p}-U_{p}$ Combine the Lagrangian function *L* and the virtual work *W*_p_ which derived above bring into Lagrange’s equation:(27)$$\;\left\{\begin{array}{c} \frac{\partial }{\partial t}\left(\frac{\delta L}{\delta {\dot{\eta }}_{\kappa }}\right)-\frac{\delta L}{\delta \eta_{\kappa }}=\frac{\delta W_{p}}{\delta \eta_{\kappa }}\\ \frac{\partial }{\partial t}\left(\frac{\delta L}{\delta {\dot{\gamma }}_{\upsilon }}\right)-\frac{\delta L}{\delta \gamma_{\upsilon }}=\frac{\delta W_{p}}{\delta \gamma_{\upsilon }} \end{array}\right.$$
where ${\dot{\eta }}_{\kappa }=\dot{q}\left(t\right),\dot{u}\left(t\right),{\dot{\gamma }}_{\upsilon }=V\left(t\right),I_{c}\left(t\right)$.

Organize and simplify Equation (27) to get the vibration equation of GPEEH, as shown in:(28)$$\left\{\begin{array}{l} M_{1}\ddot{q}\left(t\right)+T_{1}u\left(t\right)\ddot{q}\left(t\right)+T_{2}u{\left(t\right)}^{2}\ddot{q}\left(t\right)+T_{1}\dot{q}\left(t\right)\dot{u}\left(t\right)+2T_{2}u\left(t\right)\dot{u}\left(t\right)\dot{q}\left(t\right)+C_{1}\dot{q}\left(t\right)+\left(K_{1}+K_{g}\right)q\left(t\right)\\ +K_{3}q{\left(t\right)}^{3}+M_{2}g\varphi^{\prime }\left(L_{b}\right)\sin\left[\varphi^{\prime }\left(L_{b}\right)q\left(t\right)\right]u\left(t\right)+\theta V\left(t\right)=F_{y}+M_{y}\varphi^{\prime }\left(L_{b}\right)\\ -\theta \dot{q}\left(t\right)+C_{p}\dot{V}\left(t\right)+\frac{V\left(t\right)}{R_{1}}=0\\ M_{2}\ddot{u}\left(t\right)-T_{1}\dot{q}{\left(t\right)}^{2}-T_{2}\dot{q}{\left(t\right)}^{2}u\left(t\right)-M_{2}\mathrm{gcos}\left[\varphi^{\prime }\left(L_{b}\right)q\left(t\right)\right]+C_{2}\dot{u}\left(t\right)+K_{2}u\left(t\right)+\theta_{e}I_{c}\left(t\right)=0\\ -\theta_{e}\dot{u}\left(t\right)+L_{c}{\dot{I}}_{c}\left(t\right)+\left(R_{c}+R_{2}\right)I_{c}\left(t\right)=0 \end{array}\right.$$
where $M_{1}$ is the equivalent mass of GPEEH, $T_{1}$ and $T_{2}$ are the coupling terms of the modal coordinate $q\left(t\right)$ and the magnet displacement $u\left(t\right)$. $C_{1}$ is the equivalent damping, and $C_{2}$ is the spring damping. $K_{1}$ is the equivalent stiffness of GPEEH.$\;K_{3}\;$is the third-order stiffness caused by large deformation (ignore the effects of higher-order terms). *I*_2_ and *I*_p_ can calculate:$\;I_{2}=\frac{b\left(h_{b}{}^{3}-h_{a}{}^{3}\right)}{3}$,$\;I_{p}=\frac{b_{p}\left(h_{c}{}^{3}-h_{b}{}^{3}\right)}{3}$. $h_{a}=-\frac{1}{2}\frac{Eh_{1}^{2}b+2b_{p}E_{p}h_{1}h_{p}+b_{p}E_{p}h_{p}^{2}}{E_{p}b_{p}h_{p}+Ebh_{1}}$, $h_{b}=h_{1}+h_{a}$,$\;h_{c}=h_{1}+h_{p}+h_{a}$. θ is the electromechanical coupling coefficient of the piezoelectric sheet, $C_{p}$ is the equivalent capacitance of the piezoelectric sheet, $M_{2}\;$is the magnet mass, $K_{2}$ is the spring stiffness, $\theta_{e}\;$is the electromechanical coupling coefficient of the coil, and $L_{c}$ is the coil inductance, which can refer to in this paper [40]. Solving equations of these coefficients are shown in Appendix A.

Define:(29)$${\left[y_{1},y_{2},y_{3},y_{4},y_{5},y_{6}\right]}^{T}={\left[q\left(t\right),\dot{q}\left(t\right),V\left(t\right),u\left(t\right),\dot{u}\left(t\right),I_{c}\left(t\right)\right]}^{T}$$

Then, the standard form of Equation (28) can be expressed as:(30)$$\left(\begin{array}{l} dy_{1}\\ dy_{2}\\ dy_{3}\\ dy_{4}\\ dy_{5}\\ dy_{6} \end{array}\right)=\left(\begin{array}{c} y_{2}\\ \frac{\left[F_{y}+M_{y}\varphi^{\prime }\left(L_{b}\right)-T_{1}y_{5}y_{2}-2T_{2}y_{4}y_{5}y_{2}-C_{1}y_{2}-\left(K_{1}+K_{g}\right)y_{1}-K_{3}y_{1}{}^{3}-M_{2}g\varphi^{\prime }\left(L_{b}\right)\sin\left[\varphi^{\prime }\left(L_{b}\right)y_{1}\right]y_{4}-\theta y_{3}\right]}{\left[\left(M_{1}+M_{a}\right)+T_{1}y_{4}+T_{2}y_{4}{}^{2}\right]}\\ \frac{\left[\theta y_{2}-\frac{y_{3}}{R_{1}}\right]}{C_{p}}\\ y_{5}\\ T_{1}y_{2}{}^{2}+T_{2}y_{4}y_{2}{}^{2}-C_{2}y_{5}-K_{2}y_{4}-\theta_{e}y_{6}-\frac{M_{2}g\cos\left(\varphi^{\prime }\left(L_{b}\right)y_{1}\right)}{M_{2}}\\ \frac{\left[\theta_{e}y_{5}-\left(R_{2}+R_{c}\right)y_{6}\right]}{L_{c}} \end{array}\right)$$

The maximum power of piezoelectric and electromagnetic output of the energy harvester $P_{1\max}$, $P_{2\max}$, the effective output power $P_{1\mathrm{rms}}$,$\;P_{2\mathrm{rms}}\;$, the expression is as follows:(31)$$\;P_{1\max}=\frac{V_{\max}{}^{2}}{R_{1}},\;P_{2\max}=R_{2}I_{c\max}{}^{2}$$

Effective power calculating equation [42]:(32)$$P_{1\mathrm{rms}}=\frac{{\left(V_{\max}/\sqrt{2}\right)}^{2}}{R_{1}}=\frac{P_{1\max}}{2},\;\;P_{2\mathrm{rms}}=R_{2}{\left(I_{c\max}/\sqrt{2}\right)}^{2}=\frac{P_{2\max}}{2}$$

The total effective power $P_{rms0}\;$of GPEEH is as follows:(33)$$\;P_{rms0}=P_{1\mathrm{rms}}+P_{2\mathrm{rms}}$$

The calculating equation for improving efficiency η of GPEEH is as follows:(34)$$\eta =\frac{P_{rms0}-P_{crms}}{P_{crms}}\times 100\%=\left(\frac{P_{rms0}}{P_{crms}}-1\right)\times 100\%$$
where $P_{crms}$ is the effective output power of the classical galloping piezoelectric energy harvester (CGPEH).

## 3. Numerical Simulation

The model parameters of GPEEH are shown in Table 1. The electromagnetic electromechanical coupling coefficient $\theta_{e}$ is 1.12. Spring stiffness *K*_2_ is 158 N/m. Magnet quality *M*_2_ is 0.027 Kg. Then, the material properties and aerodynamic empirical coefficients in the model are shown in Table 2. By using the Runge–Kutta method of MATLAB software (MathWorks, Natick, Framingham, MA, USA) and ODE15s to solve Equation (30).

Here, the displacement of the end of the piezoelectric cantilever beam is set to $W_{1}$, the displacement of the magnet is $W_{2}$, the output voltage of the GPEH or CGPEH is expressed as Vol, and the induced current generated by the coil of EEH is expressed as $I_{c}$. The output power of the GPEH is represented by $P_{1}$, and the output power of the EEH is represented by $P_{2}$. the effective output power is expressed as $P_{\mathrm{rms}}$.

### 3.1. Optimal Matching Loads Influence of Wind on Output Characteristics

In order to study the maximum output power of GPEEH, it is necessary to analyze its optimal matching load. The effect of the matching resistance *R*_1_ of the GPEH and the matching resistance *R*_2_ of the EEH on the output performance of the GPEEH is studied. Set *R*_1_ to a fixed value (set *R*_1_ = 1 × 10^6^ Ω), change the load resistance *R*_2_, then analyze the influence of the change of *R*_2_ on the output power of the GPEH, and use the analysis results to draw the curve of the influence of wind speed U changes on the maximum output power *P*_1max_ of the GPEH, as shown in Figure 5a. Keep *R*_2_ unchanged (set *R*_2_ = 40 Ω), change the load resistance *R*_1_, analyze the effect of the change of *R*_1_ on the output power *P*_2max_ of the EEH, and use the analysis results to draw the curve of the electromagnetic maximum output power *P*_2max_ as the wind speed U changes, as shown in Figure 5b.

It can be seen from Figure 5a. that when the load resistance *R*_2_ changes in the range of 10–160 Ω, the maximum output power of the piezoelectric energy harvester increases nonlinearly with the wind speed, and the changing trend is basically the same.

When the wind speed reaches 9 m/s, the maximum difference of output power *P*_1max_ is 0.2 mW (5.2% difference). The curves of Figure 5b. have the same trend too, and the maximum difference is 0.29 mW (6.5% difference). Therefore, the mutual influence between the two can be ignored.

To specifically determine the values of the two optimal load resistance *R*_1_. Setting the load resistance *R*_2_ is 40 Ω, and study the influence of the load resistance *R*_1_ for maximum output power under different wind speeds (6, 7, 8, 9 m/s). It can be seen from Figure 6. that the maximum output power of the GPEH increases with wind speed increasing. With load resistance gradually increases, the maximum output power presents a similar normal distribution. When the load resistance *R*_1_ is 9.43 × 10^5^ Ω, *P*_1max_ of each curve reaches its peak. Thus, 9.43 × 10^5^ Ω is the optimal matching load resistance of the GPEH (*R*_1_). 

Keeping the load resistance *R*_1_ is 9.43 × 10^5^ Ω. When the wind speed is 7, 8, 8.5, 9 m/s, the output power *P*_2max_ of the EEH varies with changing of its load resistance *R*_2_, as shown in Figure 7. It can be seen from Figure 7. that when the load resistance *R*_1_ is the same, the maximum output power of the electromagnetic part *P*_2max_ increases with the increase of wind speed. When the load resistance *R*_2_, the maximum output power of the EEH increases first and the rate of increase is faster, but then gradually decreases after reaching the peak, and the changing trend becomes slower. Different wind speeds correspond to different optimal matching load resistances *R*_2_, ranging from 36–50 Ω. Because *P*_2max_ of the same curve fluctuates very little in this interval, we set the optimal electromagnetic matching load resistance *R*_2_ as 40 Ω.

### 3.2. Influence of Wind Speed on Output Characteristics

Through the above analysis, two kinds of optimally matched load resistances for GPEEH have been obtained. Next, the influence of wind speed on the output characteristics under optimally matched loads is studied, Figure 8a–c show each output parameter increases with the enhancing wind speed.

It is observed that all of the output curves in Figure 8 increase linearly or non-linearly with the rising of wind speed after reaching the start-up wind speed. In Figure 8a, the GPEH has a start-up wind speed of about 4.8 m/s. After 4.8 m/s, the curve rises rapidly and then gradually grows linearly. The maximum end displacement of the cantilever beam is 25.93 mm, and the maximum output voltage of the GPEH reaches 59.58 V. In Figure 8b, the starting point of the displacement curve of the magnet is 1.7 mm. Because the centrifugal force acts for the magnet during vibration. The starting wind speed of the electromagnetic part is 5.3 m/s and the curve grows linearly. The maximum displacement of the EEH is 21.65 mm. At this time, the maximum current output by the EEH is 10.36 mA. Figure 8b, depicts that the maximum output power of the GPEH is 3.76 mW, and the maximum output power of the EEH is 4.29 mW. 

### 3.3. Comparison between GPEEH and EEH for the Output Characteristics

The CGPEH is the structure in which the magnet is fixed but related structure and material parameters are the same as GPEEH’s. The output power of GPEEH is compared with CGPEH (*R*_1_ sets 9.43 × 10^5^ Ω and *R*_2_ is 40 Ω), as shown in Figure 9. *P*_pe_ is the total output power of GPEEH, *P*_cg_ is the output power of CGPEH, *P*_p_ is the output power of the GPEH, and *P*_e_ is the output power of the EEH.

According to these four curves in Figure 9, it can be found that *P*_cg_ is obtained by superposition of *P*_p_ and *P*_e_, so there is a superposition relationship between *P*_p_ and *P*_e_. The *P*_p_ and *P*_cg_ curves in Figure 9 do not overlap mainly because of the deviation of the solver calculation accuracy. When the wind speed is 9 m/s, the effective output power of the GPEEH reaches 3.78 mW, which is about 2.01 mW higher than that of the CGPEH of 1.71 mW. The improved efficiency η is 121%.

### 3.4. Parametric Study of EEH for Output Power

In Figure 10a, as the spring stiffness *K*_2_ gradually increases, the *P*_1max_ of GPEH first increases sharply and then remains at a constant when *K*_2_ is 200 N/m. This is mainly because when *K*_2_ is in a small range, the vibrated frequency of the cantilever beam $\omega_{p}$ gradually increases, as shown in Figure 10b, which increases the vibrated speed at the end of the cantilever beam, simultaneously the vibrated displacement increases, and finally make the output power voltage increase. However, when *K*_2_ increases to a certain value, the value of $\frac{M_{2}g}{K_{2}}$ (*M*_2_ is a fixed value) is nearly negligible in the proportion of *L*_k_ (including the spring free body constant *L*_k0_, the elongation caused by gravity $\frac{M_{2}g}{K_{2}}$), so the spring stiffness has no effect on the output power *P*_1max_.

Figure 10b is the output curves of spring stiffness *K*_2_ versus EEH maximum output power *P*_2max_. *P*_2max_ gradually decreases in non-linear trend, and finally decreases to near zero. Because the greater the rigidity, the smaller the vibrated displacement of the magnet, and the magnet hardly vibrates if the rigidity exceeds a certain level.

In Figure 10b, the output power of GPEH decreases to zero with nearly linear as the magnet mass *M*_2_ increases. This is mainly due to the longitudinal arrangement of the cantilever beam. The increase of gravity of the magnet will increase the overall gravity of the bluff body. The gravity will pull the cantilever and suppress the vibration of the cantilever. When gravity reaches a value, the bluff body does not vibrate and the system output is zero. Figure 10e. The curve shows a trend that is similar to normal distribution under different wind speeds with increasing of *M*_2_. This is mainly due to the fact that when the spring stiffness *K*_2_ is constant, with increasing of the magnet mass *M*_2_, the inertia effect enhances, and the vibrated displacement of the magnet will increase, which makes the output power *P*_2max_ increase. When *M*_2_ increases to a certain value, the *P*_2max_ reaches the peak, and then gradually decreasing cantilever beam displacement begins to affect the magnet displacement, resulting in a decrease in the magnet displacement and a decrease in the output power. Additionally, the higher the wind speed, the larger the magnet mass *M*_2_ corresponding to the peak power. Combining Figure 10b,e should try to keep in within the range of 0.025–0.03 Kg, so that both EEH and GPEH have good output characteristics.

Figure 10f,g above show the variation curves of the maximum output power of the piezoelectric part *P*_1max_ and the maximum output power of the electromagnetic part *P*_2max_ with the free elongation *L*_k0_ of the spring when the wind speed is 6, 7, 8 m/s. The curves in both figures show a parabolic-like trend and gradually decrease. The reason for the above situation is mainly due to the little decrease in the vibrated frequency of the cantilever beam $\omega_{p\;}$and the magnet $\omega_{e}$, which leads to a decrease in the vibrated displacement and a decrease in the output power. which is shown in Figure 10h. In Figure 10f, the attenuation rate little increases. In Figure 10g, the decay rate little decreases, and the higher the wind speed, the steeper the curve. At the same time, it can be seen from Figure 10h that the vibrated frequency of the magnet is twice the piezoelectric vibrated frequency.

Through the above analysis, it can be seen that the spring stiffness *K*_2_ increases, *P*_1max_ gradually increases to a fixed value and then remains unchanged, and *P*_1max_ gradually decreases to zero. In order to make GPEEH have better output characteristics, the spring stiffness *K*_2_ should be kept below 200 N/m. The effect of *L*_k0_ on the two energy harvesters is mainly because it changes the vibrated frequency of the two parts, and the larger the *L*_k0_, the lower the vibrated frequency. In addition, it was found that during the vibration process, the vibrated frequency of the magnet was twice the vibrated frequency of the cantilever beam. The magnet mass *M*_2_ is mainly due to the inertia effect and the restraining effect of gravity on *P*_2max_ and *P*_1max_, respectively. *P*_1max_ shows a gradual decrease trend until zero. *P*_2max_ is an increasing trend firstly. When the magnet mass *M*_2_ increases to a certain value, the end displacement of the cantilever beam begins to affect the magnet displacement, making it decrease. Keep the quality of the magnet within the range of 0.025–0.03 Kg, to take into account GPEH and EEH have better output characteristics.

## 4. Experimental Verification

### 4.1. Experimental Platform

To test the energy capture effect of the GPEEH and the correctness of the mathematical model, a prototype was made and the output performance test experiment of the GPEEH was carried out. The experiment equipment is shown in Figure 11. The Turbofan (DWF 3.15L, Shandong Kepuda Fan Co., Ltd., Dezhou, China) provides the wind source and filters the wind into the stable wind through the diffusion part, modulation part, and convergence part of the wind tunnel. Controler (V8 4T 4R0GB, Shenzhen Veko Technology Electronics, Ltd., Shenzhen, China) controls the wind speed of the wind tunnel. The experimental output signal collector is an oscilloscope (MDO 3014, Tektronix Inc., Beaverton, OR, USA), and the measured data can be stored in the PC using kickstart software. The base, cylinder, and bluff body are 3D printed with a resin with a density of 1140 kg/m^3^. The 3D model of overall and split GPEEH and split CPEH is shown in Figure 12. 

### 4.2. Test of Optimal Load Resistances and Maximum Output Power

The damping ratio $\xi_{1}\;$of the GPEH is 0.0089 and the damping ratio $\xi_{2}\;$of the EEH
is 0.02, measured by the free vibration attenuation method. Keeping the wind speed at 7.5 m/s, Figure 13. changes the load resistance to obtain the relationship of the curve between the load resistances and the output power of the energy harvester and compares it with the numerical analysis results. In Figure 13a, it can be seen that the optimal load resistance *R*_1_ of the GPEH is around 9.43 × 10^5^ Ω, where the output power is the largest.

The experimental test results and the numerical analysis results have the same trend. The maximum output power *P*_1max_ measured by the GPEH experiment is 1.67 mW, which is lower than the results of numerical analysis, mainly due to a certain amount of energy loss during the vibration of the cantilever beam. It can be seen from Figure 13b that the optimal load resistance value of the EEH is 46 Ω which is between 36 Ω and 50 Ω, and the experimental results of the output power changing with the load resistance are consistent with the numerical analysis results. When the wind speed is 7.5 m/s, the maximum output power *P*_2max_ of the EEH measured is 1.16 mW, which is also lower than the numerical analysis result. It is a fact that the friction force in the guide column reduces the magnet displacement during the magnet vibration, thereby reducing the output power *P*_2max._

### 4.3. Test of Comparison between GPEEH and EEH 

When the wind speed is 6 m/s, 7 m/s, and 9 m/s, the experimental test (*R*_1_ is 9.43 × 10^5^ Ω, *R*_2_ is 40 Ω) obtains the output voltage *Vol* of the GPEH and the output current *I*_c_ of the EEH as a function of time, and compare them with the numerical analysis results, as shown in Figure 14. It can be seen from Figure 14 that the changing trend of the experimental results is basically consistent with the changing trend of the numerical analysis results. When the steady-state limit cycle oscillation is reached, the amplitude of the data is basically the same. It can be found from the experimental results that $\omega_{p}$ the vibrating frequency of the piezoelectric part is nearly 1/2 of $\omega_{e}$ the vibrating frequency of the electromagnetic part ($\omega_{p}\;$are 3.8 Hz, 4.03 Hz, 3.93 Hz and$\;\omega_{e}\;$is 7.7 Hz, 8.13 Hz, 8.2 Hz, when U are 6 m/s, 7 m/s, 9 m/s, respectively). Figure 15a shows the output voltage of the CGPEH and GPEEH when the load resistance *R*_1_ = 9.43 × 10^5^ Ω and *R*_2_ = 40 Ω. The experimental test results and the numerical analysis results of output current *I*_cmax_ is compared as shown in Figure 15b. It can be seen from Figure 15a that the experimental results measured and the numerical analysis results non-linearly promotion with the increase of the environmental wind speed, and the changing trend is the same. The initial vibrating wind speed of the GPEH is 5 m/s, which is lower than the starting wind speed of CGPEH of 5.5 m/s; In the range of 5–7.6 m/s, and the output voltage of GPEEH is higher than that of CGPEH, and numerical simulation who calculated range of 4.8–8.08 m/s has a small deviation.

Figure 15b shows the comparison between the experimental test results and the numerical analysis results of the output current of the EEH. It can be seen that the trend of experimental test results and the numerical analysis results are consistent. In the theoretical calculation, we ignored the influence of resistance on the system for the convenience of research, but in the experiment, the guide rod and the bearing could not be absolutely smooth. This is the reason why the EEH experiment result is lower than the simulation result.

Figure 15c shows the comparison diagram of GPEEH and CGPEH effective output power between experiment and simulation. During the experiment, the GPEEH output voltage and current waveforms are usually unstable, but in order to simplify the research process, we also use Equation (32) during calculating the effective power of the experimental part. The values and trends of experiments and simulations of GPEEH and CGPEH in Figure 15c are the same. At 9 m/s, the effective output power of the GPEEH experiment reached 3.57 mW. The effective output power of the CGPEH experiment reached 1.68 mW. The experimental improvement efficiency was 112.5%, which was 8.5 % worse than that of the simulation result. In summary, the simulation and experiment are consistent and the deviation is within the acceptable range, so the correctness of the model is verified. 

## 5. Conclusions

This paper proposes a galloping piezoelectric–electromagnetic energy harvester (GPEEH), designs a theoretical model, deduced vibration coupling equation, and analyzes the influence of wind speed U, load resistances *R*_1_, *R*_2_, and key parameters of EEH for the output performances. The main conclusions are as follows:

(1) The method of numerical analysis determines the best matching load resistance of the system. On this basis, the influence of wind speed on the output characteristics of GPEH and EEH is studied. Numerical simulation results show that the starting wind speed of GPEH is about 4.8 m/s, and the starting wind speed of EEH is 5.3 m/s; The effective output power of GPEEH is 3.78 mW, and the effective output power of CGPEH 1.71 mW. The improved efficiency η is 121%

(2) The study for output performances of parameters of EEH include: Spring stiffness *K*_2_, Magnet quality *M*_2_, and Free extension of spring *L*_k0_. With increasing of *K*_2_, *P*_1max_ first fast increases to a fixed value and then remains unchanged, and *P*_1max_ gradually decreases to zero. The influence of *L*_k0_ on the output power of GPEH and EEH is mainly due to the change of the vibrated frequency. The larger the *L*_k0_ is, the smaller the vibration frequency is, the smaller the output displacement is, and the smaller the output power is. *M*_2_ is the effect of gravity and inertia on the output power *P*_1max_ and *P*_2max_, respectively. *M*_2_ gradually increases, *P*_1max_ gradually decreases until it reaches 0, and *P*_2max_ first increases and then decreases. We can get from the figure, try to control *K*_2_ below 200 N/m and *M*_2_ between 0.025 and 0.03 Kg, so that GPEEH has great output characteristics.

(3) The wind tunnel was verified by experiments and found that the experimental results are consistent with the simulation results. When the wind speed is 9 m/s, the effective output power of GPEEH is 3.57 mW, and CGPEH reaches 1.68 mW. The effective power increased by 112.5%. The deviation is within an acceptable range, verifying the correctness of this model.

## Figures and Tables

**Figure 1 micromachines-12-00626-f001:**
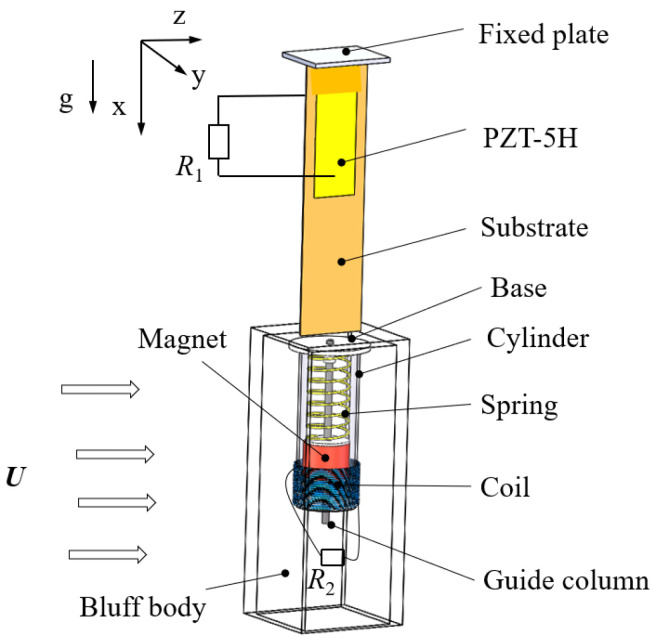
The structure of GPEEH.

**Figure 2 micromachines-12-00626-f002:**
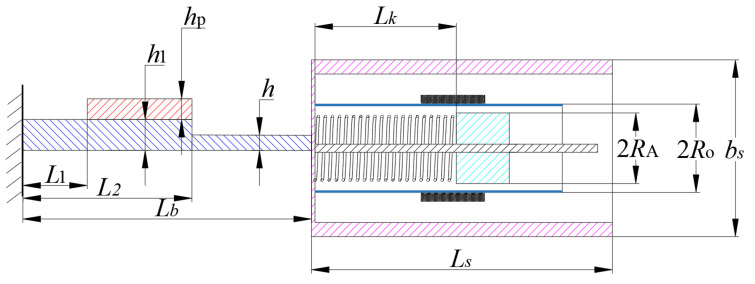
Cross-section diagram of GPEEH.

**Figure 3 micromachines-12-00626-f003:**
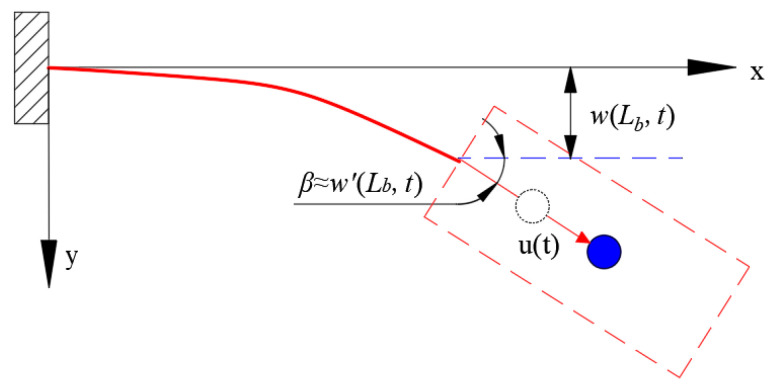
Deformation diagram of the cantilever beam.

**Figure 4 micromachines-12-00626-f004:**
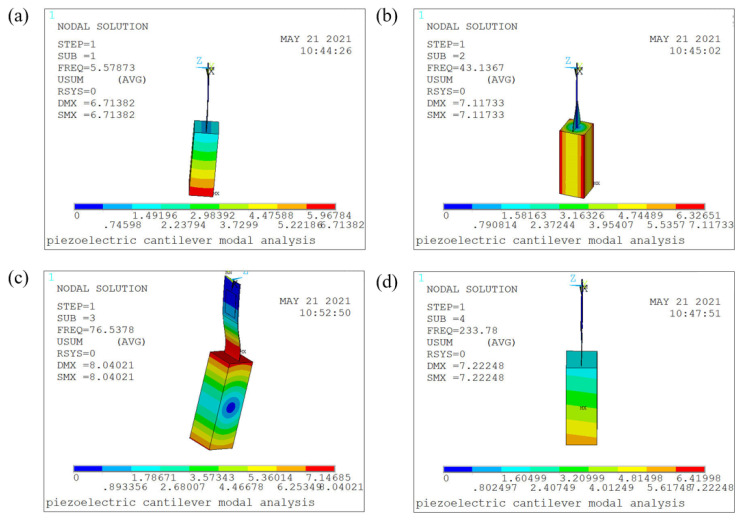
Stress cloud diagram of GPEEH modal analysis. (**a**) First mode (**b**) Second mode (**c**) Third mode (**d**) Fourth mode.

**Figure 5 micromachines-12-00626-f005:**
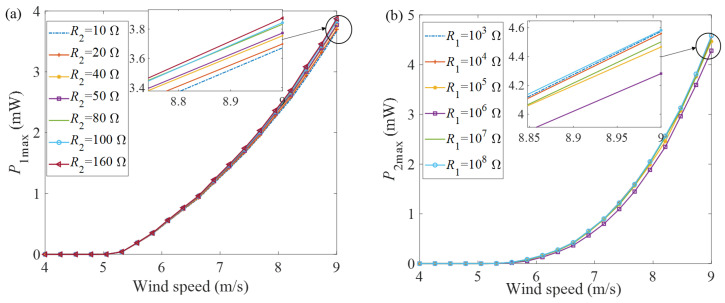
Curves of maximum output power with wind speed under different loads (**a**) *P*_1max_ (**b**) *P*_2max_.

**Figure 6 micromachines-12-00626-f006:**
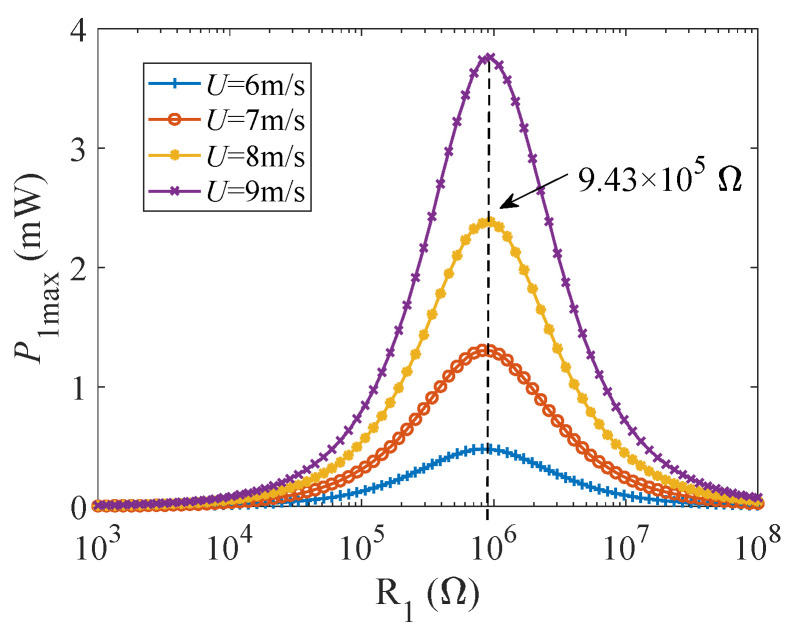
The curve diagram of the maximum output power *P*_1max_ of GPEH with its load resistance *R*_1_ (U = 6, 7, 8, 9 m/s).

**Figure 7 micromachines-12-00626-f007:**
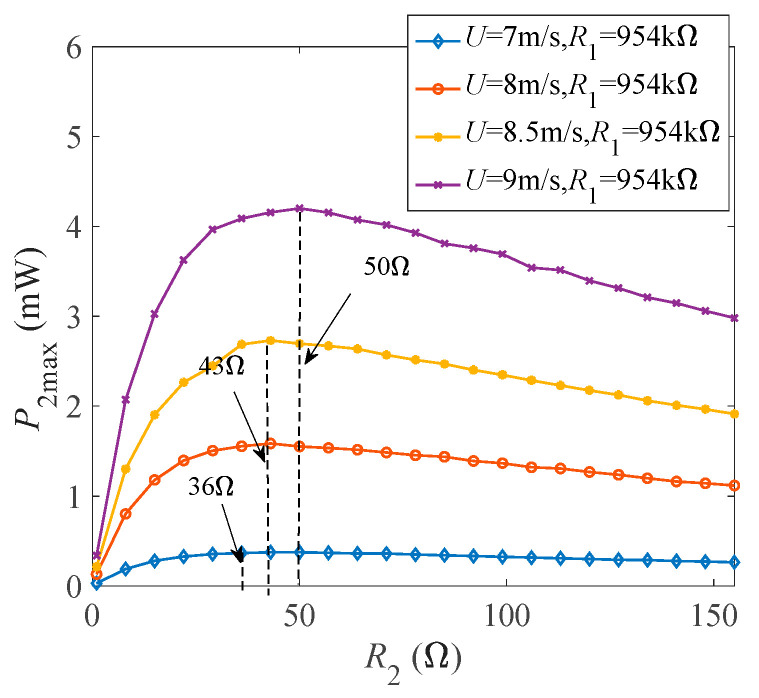
The curve of the output power *P*_2max_ with its load resistance *R*_2_ (U = 7, 8, 8.5, 9 m/s).

**Figure 8 micromachines-12-00626-f008:**
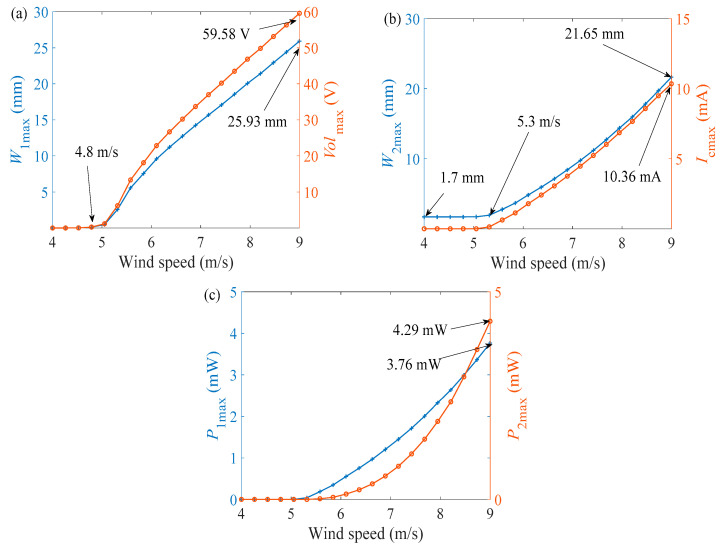
Curves of output parameters changing with wind speed under optimally matched loads (**a**) *W*_1max_ and *Vol*_max_ (**b**) *W*_2max_ and *I*_cmax_ (**c**) *P*_1max_ and *P*_2max_.

**Figure 9 micromachines-12-00626-f009:**
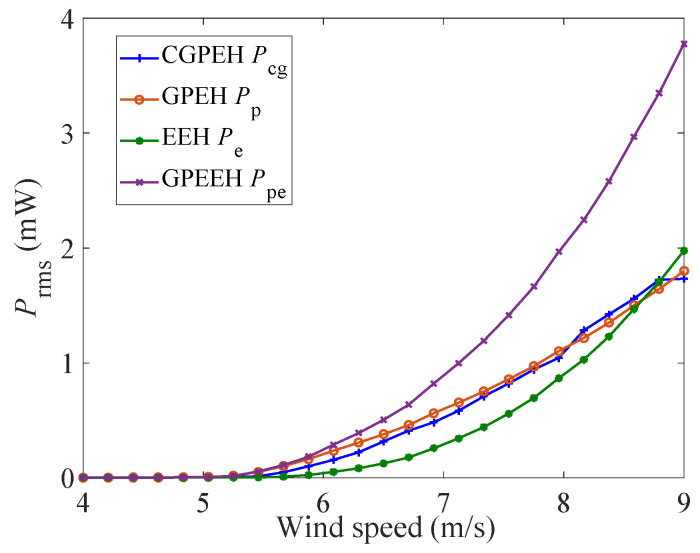
Comparison of effective output power between GPEEH and CGPEH.

**Figure 10 micromachines-12-00626-f010:**
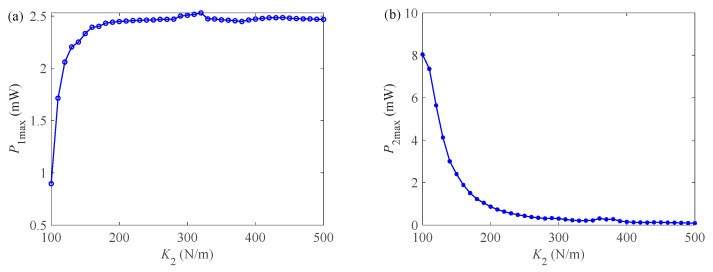
The influence of EEH parameters on output characteristics. (**a**) *P*_1max_ with increasing *K*_2_. (**b**)*P*_2max_ with increasing *K*_2_. (**c**) $\omega_{p}$ with increasing *K*_2_. (**d**) *P*_1max_ with increasing *M*_2_. (**e**) *P*_2max_ with increasing *M*_2_. (**f**) *P*_1max_ with increasing *L*_k0_. (**g**) *P*_2max_ with increasing *L*_k0_. (**h**) The vibrated frequency with increasing *L*_k0_.

**Figure 11 micromachines-12-00626-f011:**
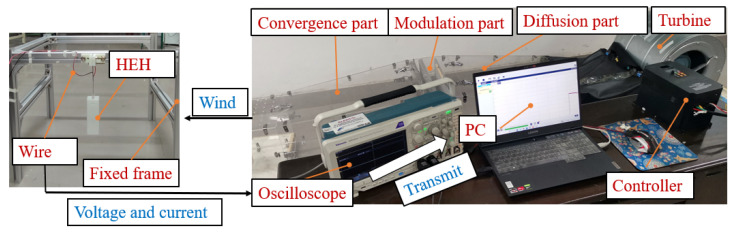
Schematic diagram of experiment equipment.

**Figure 12 micromachines-12-00626-f012:**
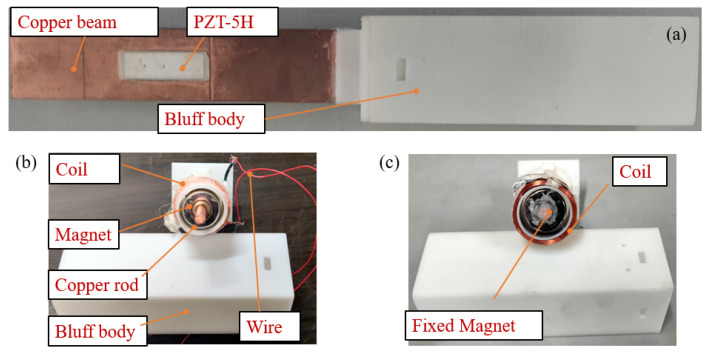
(**a**) The overall GPEEH (**b**) The split GPEEH (**c**) The split CGPEH.

**Figure 13 micromachines-12-00626-f013:**
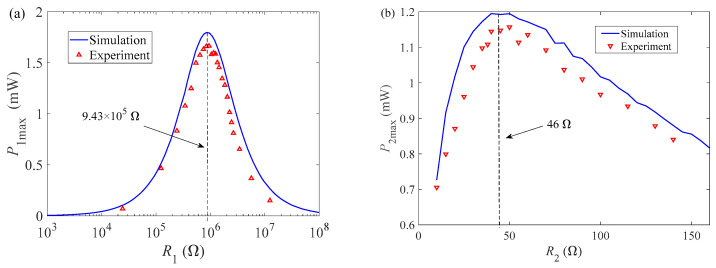
Output power curve with the load resistances. (**a**) Comparison between *R*_1_ optimal load experimental value and numerical analysis value (**b**) *R*_2_ optimal load experimental value and numerical analysis value comparison.

**Figure 14 micromachines-12-00626-f014:**
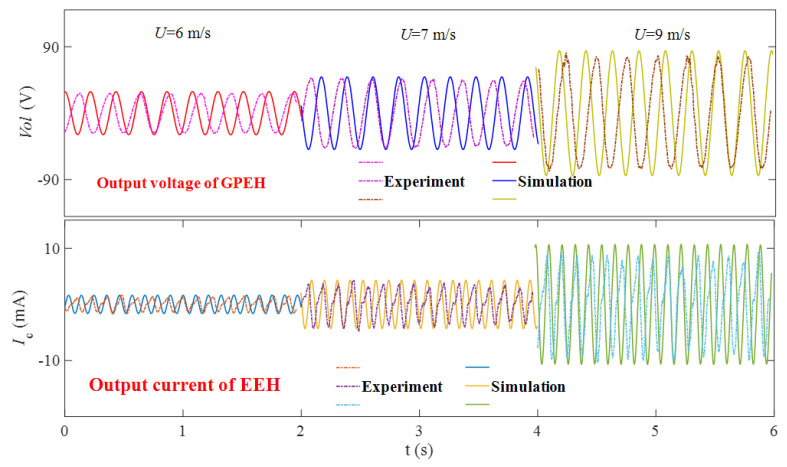
Comparison of time-domain diagram of experimental results and numerical analysis results of the output performance of GPEEH at 6, 7, 9 m/s.

**Figure 15 micromachines-12-00626-f015:**
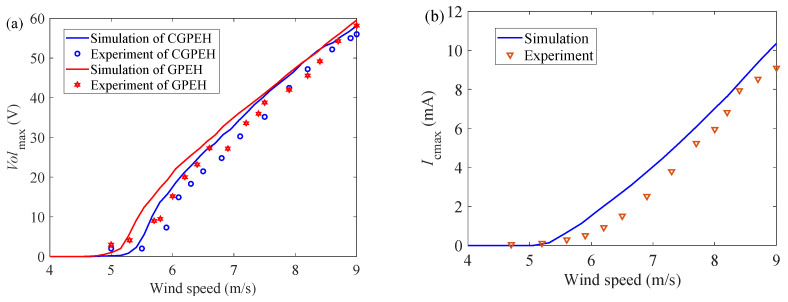
GPEEH and CGPEH output parameters of experimental and simulated change with wind speed (**a**) *Vol*_max_ (**b**) *I*_cmax_ (**c**) *P*_rms_.

**Table 1 micromachines-12-00626-t001:** Structure parameters.

Parameter	Symbol	Value
Length, width, and height of the front substrate,	$L_{b},b,h_{1}$(mm)	114, 30, 1.08
Length, width, and height of the rear substrate,	$L_{b},b,h$(mm)	114, 30, 0.5
Length, width, and height of PZT-5H,	$L_{p},b_{p},h_{p}\;$(mm)	32, 10, 0.2
Height, length, and width of the bluff body,	$L_{s},b_{s},b_{s}$(mm)	120, 40, 40
Coil inside, outside radius and height,	$R_{i},R_{o},L_{c}$(mm)	13, 15,1 8.6
The magnet inside, outside radius, and height,	$R_{A},R_{B},L_{m}\;\left(\mathrm{mm}\right)$	5, 10, 15
Copper rod height,	*L*_g_ (mm)	70
Cylinder height,	*L*_t_ (mm)	90
Number of coil turns,	n	792
Free extension of spring,	$L_{k0}$(mm)	51

**Table 2 micromachines-12-00626-t002:** Material property parameters and other correlation coefficients.

Parameter	Symbol	Value
Cross-sectional area,	$A,A_{p}\;$(m^2^)	1.5 × 10^–5^, 2 × 10^–6^
Density,	$\rho ,\rho_{p}\;$(kg/m^3^)	8950, 7500
Young’s modulus,	$E,E_{p}\;$(GPa)	110, 60.6
PZT dielectric constant,	$\varepsilon_{33}^{s}\;$(nF/m)	25.6
PZT piezoelectric constant,	$e_{31}\;$(C/m^2^)	−16.6
Acceleration of gravity	g (m/s^2^)	9.8
Aerodynamic experience coefficient [20]	$a_{1},a_{3}$	2.3, −18

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
