# Peer review of "A Piezoelectric and Electromagnetic Hybrid Galloping Energy Harvester with the Magnet Embedded in the Bluff Body"

_micromachines, 2021, doi:10.3390/mi12060626_

Round 1

Reviewer 1 Report

The work presented in this manuscript is about a cantilever-type energy harvester that is driven by wind flow passing through the harvester's free-end buff body.  The presentation of the manuscript is logical and easy to follow.

The authors did a very good in defining the scope of the proposed research and device supported by sufficient literature reviews.  Since the idea proposed in this work is kind of utilizing the flow-induced vibrations of flexible plate in the wake of square cylinder, it is highly suggested the  literature https://www.mdpi.com/1996-1073/13/10/2645 to be included in this work.

Since the device is based on the vibrations of the plate, it is suggested that perhaps the modal analysis of the plate to be addressed to highlight which vibratory mode of the plate dominates the generation of the collected energy.

The English of the present work is fine and easy to understand.  However there is a grammatical error in line 28-29, the first sentence in the Introduction section that reads as "To solve the difficulty of the microelectronic device[1] to replace the battery, and environmental pollution caused by used batteries[2]. "  Basically this is not a sentence at all as it does not contain any verb! 

In general, the manuscript is acceptable for publication expect the two points mentioned above.

Reviewer 2 Report

This paper studied the characteristics of a galloping piezoelectric-electromagnetic energy harvester (GPEEH) theoretically and experimentally. First, theoretical model was designed, the vibration coupling equation was deduced and the influences of some parameter for the output performances were analyzed. Then, the experiment using a wind tunnel was conducted and the experimental results are compared with theoretical results. I would like to forward the following comments to the authors:

  1. The author should explain whether the combination of piezoelectric and electromagnetic parts showed a synergistic effect or was just a superposition.
  2. In Fig. 14, if there is a reason why simulation result of CGPEH show non-smooth line, the author should explain the reason. And, in line 555, what does (42) indicate?
  3. I can find some mistakes. Such as, the strange sentence (line 155), the unnecessary period (line 174, line 193), error the equation (Eq. (34)), the fluctuation of unit (Fig. 9 (d) and (e)).

In short, I suggest that this paper would be accepted for publication after the authors addressed the above questions and/or revisions.

Reviewer 3 Report

Authors proposed galloping piezoelectric-electromagnetic energy harvester (GPEEH) which is interesting device. Authors used mathematical model and simulated results to prove the output power of GPEEH which is higher than galloping piezoelectric energy harvester. Figures and Tables are very clear to be seen. The mathematical analysis is in detail to understand the concept. Overall, the manuscript is well written. However, there are some missing references. Therefore, the manuscript could be minor revision if authors follow the suggestions.

1.  Please increase label sizes of Figures 1 and 2.
2.  Please use clearer font in Figures 10 and 11.
3.  In Figure 14, are there any saturation points ?
4. In Figure 13, authors might provide the frequency response data if possible.
5. Please add some information in Author contribution, Funding, Acknowledgments, conflicts of interest section.
6. In Line 677, Ieee -> IEEE.
7. Please provide the reference for the sentence (The energy conversion mechanism of PEH is to convert the vbration energy of the environmental vibration energy into the strain energy of piezoelectric ceramic.  ) with the reference ( Sodano, Henry A., Daniel J. Inman, and Gyuhae Park. "A review of power harvesting from vibration using piezoelectric materials." Shock and Vibration Digest 36.3 (2004): 197-206.) or another reference.
8.Please provide the reference for the sentence (Based on the positive piezoelectric effect, positive and negative charges are generated on the upper and lower surfaces of the piezoelectric plate,  ) with the reference (Kim, J.; You, K.; Choe, S.-H.; Choi, H. Wireless Ultrasound Surgical System with Enhanced Power and Amplitude Performances. Sensors 2020, 20, 4165.).
9.Please provide the reference for the sentence ( Galloping generally happens on a complex, irregular, and non-streamlined cross-sectional structure with edges a galloping energy harvester has usually a single degree of freedom, as the spring-damping-mass model ) with the reference (Zhao, Kaiyuan, Qichang Zhang, and Wei Wang. "Optimization of galloping piezoelectric energy harvester with V-shaped groove in low wind speed." Energies 12.24 (2019): 4619. ) or another reference.
10. In line 483, In order to -> To.
11. Please use formal English expression (And -> In addition in Line 51).
